# Application of Sugar Beet Pulp Digestate as a Soil Amendment in the Production of Energy Maize

Andrzej Baryga [1], Bożenna Połeć [1], Andrzej Klasa [2,*] and Tomasz Piotr Olejnik [3]

[1] Prof. Wacław Dąbrowski Institute of Agricultural and Food Biotechnology, Ul. Rakowiecka 36, 02-532 Warszawa, Poland; andrzej.baryga@ibprs.pl (A.B.); bozenna.polec@ibprs.pl (B.P.)

[2] Department of Agricultural and Environmental Chemistry, University of Warmia and Mazury in Olsztyn, Ul. Oczapowskiego 8, 10-519 Olsztyn, Poland

[3] Institute of Food Technology and Analysis, Lodz University of Technology, Ul. B. Stefanowskiego 4/10, 90-924 Łódź, Poland; tomasz.olejnik@p.lodz.pl

* Correspondence: aklasa@uwm.edu.pl; Tel.: +48-523-4705

**Abstract:** This study aimed to determine the suitability of sugar beet pulp digestion by-products as soil amendments for maize grown for energy purposes. In a plot experiment, nitrogen fertilizer at a standard rate of 200 kg N ha$^{-1}$ was applied as a control vs. treatment with solid and liquid digestate fractions. Digestate was obtained from a gasifier processing only sugar beet pulp. Following harvest, heating and calorific values were determined together with the yield and chemical composition of the maize cob and stover. It was found that soil amendment with crude (unseparated) digestate or its two fractions (separated into liquid and solid) produced higher yields of maize cobs and stover compared with the application of standard fertilizer. An analysis of the energy value of the maize plant revealed differences between the studied soil treatments. Cobs obtained from plots treated with the digestate showed higher calorific and heating values than those obtained from control plots; however, maize stover from control plots showed higher calorific and heating values compared with plants from other experimental plots. It can be concluded that by-products obtained from sugar beet pulp digestion can be alternatively used as a soil amendment for maize production in a crop rotation with sugar beet. Among studied amendments the solid fraction of the digestate was found to have the best performance.

**Keywords:** sugar beet pulp digestion; digestate; energy maize; yield; biomass calorific and heating values





## 1. Introduction

Although sugar beet pulp has served as a traditional valuable feed for dairy cows, due to changes in agricultural systems and economic factors in various countries, alternative utilization methods of this by-product have been introduced. Because of its physico-chemical properties, sugar beet pulp appears to be an excellent stock material for anaerobic digestion [1–6]. Under optimal conditions, biogas obtained from sugar beet pulp contains (in % vol.): 52–85% of methane, 14–48% of carbon dioxide, 0.08–5.5% of hydrogen sulfide, 0.0–5.5% of hydrogen, 0.0–2.1% of carbon monoxide, 0.6–7.5% of nitrogen and 0.0–1.0% of oxygen. The biogas composition is determined by the stock material and technology applied in a biogas plant [2,4,7].

The process of anaerobic digestion is characterized by the generation of an intrinsic by-product called digestate. Digestate is a mixture of solid and liquid fractions. Extensive studies have been conducted worldwide to define the economically and environmentally optimal digestate utilization method [8–11]. It was concluded that the application of digestate into the soil is a suitable solution. This method reduces ground and surface water pollution by nutrients. Moreover, the high concentration of nutrients, which are contained in the digestate and are available to plants, can limit the consumption of fertilizers that have a significant impact on farm economics [12].

Based on a previous publication of the authors of the present study [13–18], at least a fraction of the fertilizers applied to sugar beet fields can be successfully replaced by nutrients contained in digestate from sugar pulp gasification. It is worth mentioning that fertilizers represent a considerable portion of the structure of production costs (for sugar beet production, the costs of fertilizers amount to 27% of the total production costs). The replacement of mineral fertilizers by by-products can also have a wide environmental context because some raw materials used in the fertilizer industry are contaminated by harmful elements such as cadmium or uranium and the use of alternative sources of nutrients can lead to reduction of soil contamination [19].

Finding other crops that could use the nutrients of the digestate would be especially beneficial for farmers interested in the utilization of digestate from agricultural gasification plants.

According to the authors, the best candidate seems to be maize (*Zea mays*), a crop with multiple applications in agricultural systems. It is a food and fodder crop of high efficiency [20,21]. Due to the high rate of biomass accumulation through physiological mechanisms, maize has gained a high interest as a crop grown for energy purposes [22–24]. Although maize has been grown all over the world as a feedstock for liquid biofuel and biogas [25], maize residues can also be used for the generation of heat energy [26,27]. To produce maize biomass more profitably, organic wastes can be applied. This is allowed since maize grown for energy may be treated as a non-food and non-fodder crop and, in many regions of the world, legal regulations provide for the utilization of organic waste. Maize is characterized by high yielding ability, high carbohydrate content and wide applicability in the renewable energy sector. Maize biomass can be used for direct combustion and production of liquid biofuels [28–32]. The calorific value of the direct combustion of maize stover and cob (or grains) amounts to 15 and 19 GJ $Mg^{-1}$, respectively.

To obtain a high yield of maize biomass, appropriate amounts of nutrients have to be supplied to the soil to cover the high demands of this crop [33,34]. The nutritional demands for maize grown for anaerobic digestion of whole-crop silage reported by [33] are as follows: nitrogen 130–260 kg N $ha^{-1}$; phosphorous 23–46 kg P $ha^{-1}$ (53–106 kg $P_2O_5$ $ha^{-1}$); and potassium 131–262 kg K $ha^{-1}$ (157–314 kg $K_2O$ $ha^{-1}$) for a predicted yield of fresh biomass of 100–200 Mg $ha^{-1}$, respectively.

Under the standard conditions of Polish agriculture, the yield of maize grain ranges from 5 to 10 Mg $ha^{-1}$ and for cob and stover 8 to 20 Mg $ha^{-1}$, respectively. When considering the cost structure of maize production for silage or grain, the cost of materials, including seeds, pesticides and fertilizers amounts to 51% for silage and 60% for grain [35].

In addition to the nutrient supply, the pattern of weather conditions can determine the yield of maize biomass. High moisture content in the soil and a relatively high air temperature following sowing are favorable for rapid plant emergence and the development of plant canopy. It was found that the precipitation pattern in summer is a crucial factor. Precipitations higher than 107, 151 and 75 mm $m^{-2}$, in July, August and September, respectively, negatively affect the health status of the crop, make the growing season longer and causing low yields of maize grain [36].

Other Polish authors [20,37] reported that, in the growing months, precipitation levels higher than 350–400 mm $m^{-2}$ resulted in a considerable reduction in grain yield, especially when the temperature was relatively low (ca. 14 °C).

This study aimed to verify whether the application of sugar beet pulp digestate is a suitable practice in maize cultivation and how it impacts the yield and energy parameters of maize biomass.

## 2. Materials and Methods

The study materials included maize biomass (*Zea mays* cv. Cannavaro) obtained from experimental plots and sugar beet pulp digestate in three forms: unseparated, liquid fraction and solid fraction. Sugar pulp digestate originated from the experimental installation that has been working at the Institute of Agricultural and Food Biotechnology since 2012

where sugar pulp from Polish sugar beet processing plants was digested. A continuous system of feeding the pulp to the digester was used with a peristaltic pump.

A plot experiment was performed at the Experimental Center in Leszno. The area of a single plot was 18.75 m$^2$ each and the plots were separated by panels (polyethylene-laminated particleboard). Latin square design was applied because of lack of soil variability. The forecrop in each season of study was sugar beet.

The control plots were fertilized only with mineral nitrogen fertilizer matched to the nutrient demands of maize (200 kg N ha$^{-1}$), whereas the soil of experimental plots was amended with the crude, liquid or solid fraction of digestate characterized by the same content of nitrogen in each of the forms. The studied treatments are presented in Table 1.

**Table 1.** Studied treatments.

| Type of Fertilizer | Rate in kg Plot$^{-1}$ | Rate in Mg Ha$^{-1}$ |
|---|---|---|
| NPK* (complex fertilizer "Lubofos CORN"®) | 7.5 | 4.0 |
| Liquid fraction | 35.4 | 18.9 |
| Raw digestate | 16.3 | 8.0 |
| Solid fraction | 21.6 | 11.5 |

* NPK–complex fertilizer contained nitrogen, phosphorus and potassium.

Studies were conducted in three successive growing seasons, from 2014 to 2016. Weather patterns of the studied seasons are shown in Figures 1–3. The temperature patterns of the studied seasons were very different, and the season of 2014 was the most appropriate for maize cultivation, especially in combination with a relatively high precipitation at the plant emergence phase. At the time of grain maturation, the season of 2015 was characterized by favorable temperatures, although soil drought reduced the yield.

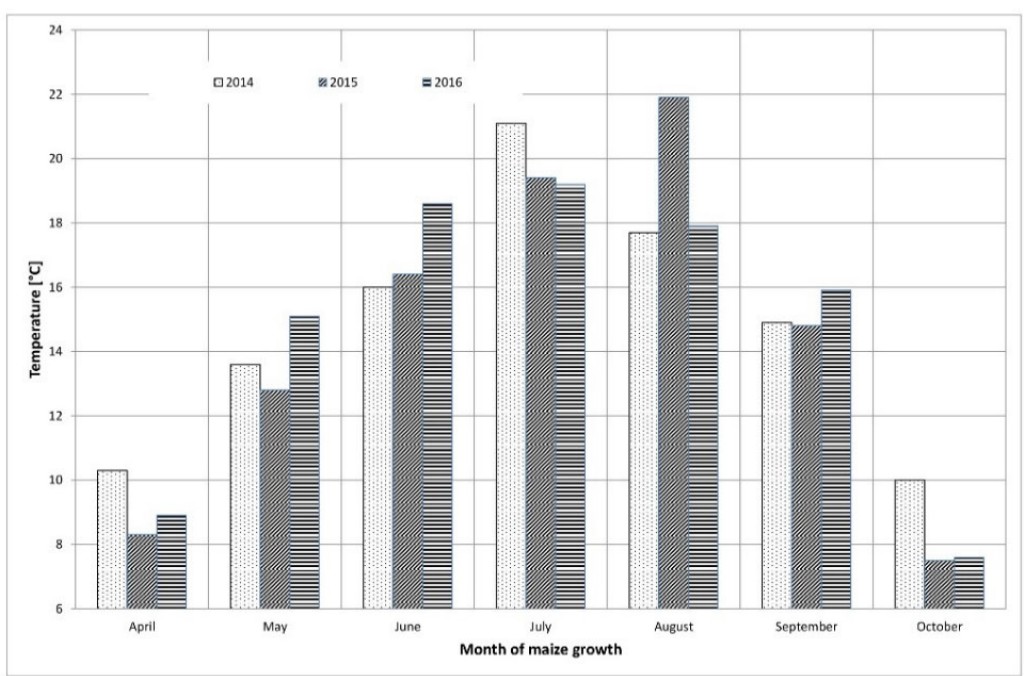

**Figure 1.** Average monthly temperature in seasons of studies 2014–2016.

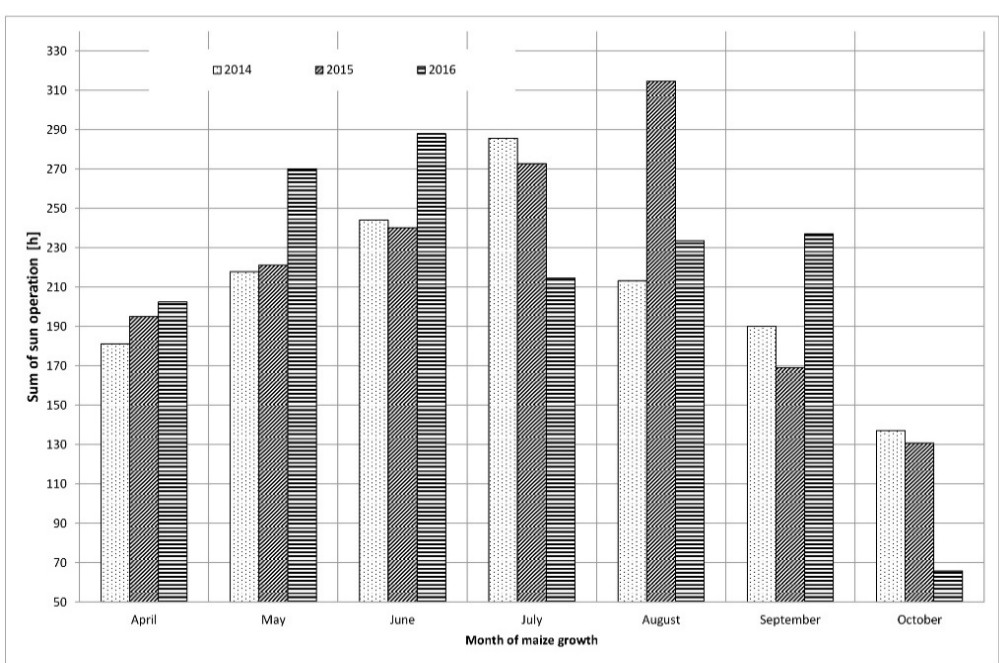

**Figure 2.** Total insolation in the growing seasons of 2014–2016 (in hours/month).

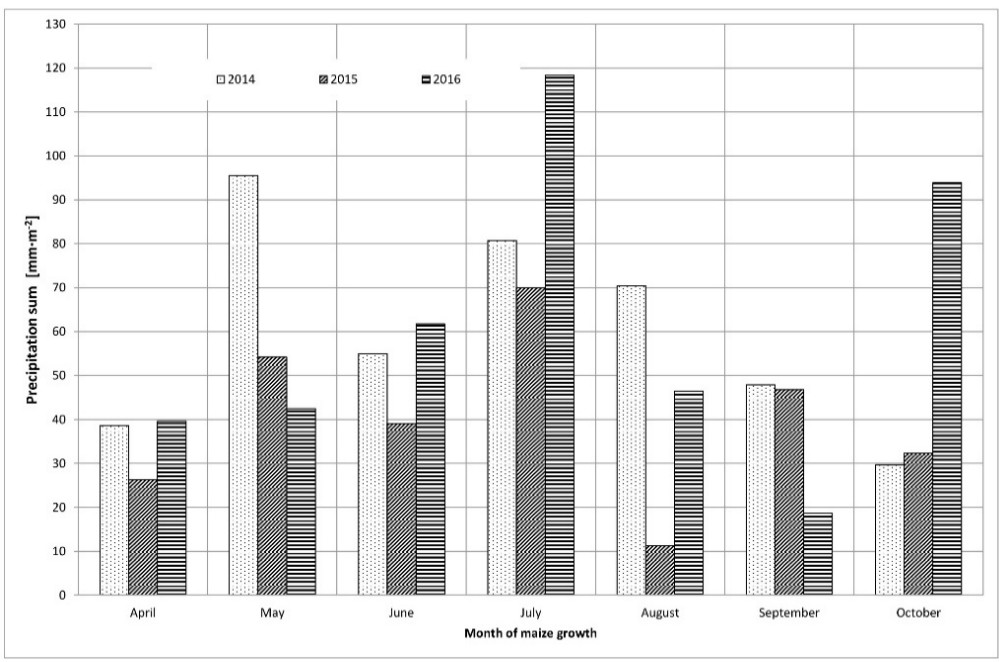

**Figure 3.** Total precipitation in the growing seasons of 2014–2016.

Following harvest of the fully mature crop, the following analyses were performed (number of European laboratory standard is given in brackets): total moisture content—gravimetric method (Q/LP/05/A:2011); ash content—gravimetric method (Q/LP/06/A:2011); volatile fraction—gravimetric method (Q/LP/07/A:2011); hydrogen and nitrogen content—high-temperature combustion method (Q/LP/09/B:2012); total sulfur content—high-temperature combustion method (Q/LP/08/A:2011); GCV (Gross Calorific Value)—calorimetry (Q/LP/12/A:2011); NCV (Net Calorific Value) by calculation (Q/LP/12/A:2011).

To determine the chemical quality and energy value of the maize biomass, the methodology used for solid fuel was applied and the respective standard values were compared with crude biomass, i.e., the biomass at the harvest.

Statistical analyses were performed using two-way ANOVA at $p = 0.05$. The significance of differences was calculated using Fisher's test. The software used was Statistica ver. 12.0 PL.

The digestate and its fractions were analyzed using the following procedures: pH—ionomeric electrode (PN-EN 12176:2004); dry matter content—gravimetric method (PN-EN 12880:2004); total organic substances (PN-EN 12879:2004); content of cadmium, copper, nickel, lead and zinc—FAAS, (PN-EN 13346:2002 8.3 and PN-ISO 8288:2002); chromium—(FAAS), PN-EN 13346:2002 8.3 and PN-EN 1233:2000 3; mercury—CV-AAS (2, 2014-EN 13346:2002 8.3 and PN-EN 1233:2000 3); calcium and magnesium—FAAS (PN-EN 13346:2002 8.3 and PN-EN ISO 7980:2000); total nitrogen—Kjeldahl's titration (PN-EN 13342:2002); total phosphorus—spectrophotometry (PB-84 edition 2 201); potassium—FAAS (PB-PAC-03, edition 2, 2015); the presence of salmonella and living eggs of gut parasites—*Ascaris* sp., *Trichuris* sp., *Toxocara* sp.—PN-EN ISO 6579:2003/A1:20.

## 3. Results

### 3.1. Analyses of Digestate Obtained from Sugar Beet Pulp Anaerobic Digestion

Results of chemical analyses of digestate samples from the sugar beet pulp processing plant and the element threshold levels regulated by law are shown in Table 2.

Based on the presented results, it can be concluded that the amounts of nitrogen and phosphorous were higher in the digestate solid fraction compared with that in the unseparated digestate or its liquid fraction, whereas the potassium concentration in the unseparated digestate was higher than that in the other two experimental forms of this soil amendment.

When considering the essential nutrient content, the chemical properties of the digestate used in this study, are in line with the data of available literature on the characteristics of "typical" digestates obtained from installations processing agricultural and food wastes [38–40]. The cited authors studied the chemical properties of both fractions of digestate and found that the values of the parameters under study ranged considerably, i.e., the total nitrogen content ranged from 0.4% to 0.8% and from 0.29% to 0.75% in the solid and liquid fractions, respectively. Similar variabilities were reported for potassium and phosphorous.

It should be mentioned that digestate can be characterized by an abundance of other nutrients such as calcium (87–131 g kg $DM^{-1}$), magnesium (3.6–11.1 g kg $DM^{-1}$) and boron (0.19–0.22 g kg $DM^{-1}$).

Although in the studied digestate forms, trace amounts of cadmium, lead, mercury, copper, cobalt and zinc were determined, in each case, their content was considerably lower compared with the permissible limit established for the application of waste for soil amendment.

The studied digestate and its fractions did not contain any salmonella or any living eggs of gut parasites, which is indicative of the anaerobic digestion's effectiveness in ensuring microbiological safety.

When comparing the chemical parameters of the studied digestate with the respective values of permitted thresholds (provided for in the regulation of the Minister of Agriculture and Rural Areas Development) [41], the experimental digestate and its fractions can be applied in soil without any restrictions, irrespective of the purpose of the produced biomass (food, feed or energy biomass).

**Table 2.** Comparison of the quality of the experimental digestate against the permissible thresholds.

| Parameter | Parameters of Used By-Products from Anaerobic Digestion: | | | | | | | | | Permitted Level Mg kg DM$^{-1}$ or Number, Respectively |
|---|---|---|---|---|---|---|---|---|---|---|
| | 2014 | | | 2015 | | | 2016 | | | |
| | Digestate | Liquid Fraction | Solid Fraction | Digestate | Liquid Fraction | Solid Fraction | Digestate | Liquid Fraction | Solid Fraction | |
| Acidity [pH-H$_2$O] | 7.5 | 7.4 | 9.0 | 7.6 | 7.5 | 9.2 | 7.7 | 7.5 | 9.8 | – |
| Dry matter (DM) [% FW] | 2.5 | 0.3 | 92.0 | 2.3 | 0.5 | 93.6 | 4.0 | 0.5 | 91.1 | – |
| Organic substances [% DM] | 51.6 | 56.2 | 89.0 | 50.9 | 62.0 | 93.4 | 56.8 | 61.6 | 88.5 | – |
| Cadmium (Cd) | 2.20 | 2.05 | 2.20 | 5.20 | 3.14 | 2.78 | 1.23 | 2.19 | 2.76 | ≤20 |
| Lead (Pb) [mg kg DM$^{-1}$] | 42.4 | 29.8 | 36.7 | 22.1 | 32.7 | 20.1 | <25 | 42.4 | 15.4 | ≤750 |
| Nickel (Ni) [mg kg DM$^{-1}$] | 8.8 | 7.6 | 6.80 | 5.5 | 5.4 | 5.13 | 3.81 | 12.8 | 5.03 | ≤300 |
| Chromium (Cr) [mg kg DM$^{-1}$] | 25.0 | 25.0 | 25.0 | 26.3 | 25.0 | 25.0 | 28.4 | 25.0 | 29.9 | ≤500 |
| Mercury (Hg) [mg kg DM$^{-1}$] | 0.357 | 0.240 | 0.350 | 0.426 | 0.325 | 0.432 | 0.50 | 0.357 | 0.555 | ≤16 |
| Copper (Cu) [mg kg DM$^{-1}$] | 88 | 76 | 98 | 115 | 78 | 112 | 150 | 88 | 110 | ≤1000 |
| Zinc (Zn) [mg kg DM$^{-1}$] | 295 | 310 | 325 | 470 | 380 | 498 | 366 | 295 | 882 | ≤2500 |
| Calcium (Ca) [g kg DM$^{-1}$] | 112 | 84 | 112 | 134 | 96 | 148 | 77 | 82 | 132 | – |
| Magnesium (Mg) [g kg DM$^{-1}$] | 8.4 | 6.92 | 2.89 | 11.4 | 7.48 | 4.12 | 13.5 | 8.35 | 3.74 | – |
| Total nitrogen (N) [g kg DM$^{-1}$] | 23.0 | 10.6 | 17.4 | 20.7 | 10.8 | 27.6 | 24.6 | 11.4 | 29.4 | – |
| Total phosphorus (P$_2$O$_5$) [g kg DM$^{-1}$] | 1.59 | 1.00 | 2.20 | 1.26 | 1.68 | 3.60 | 1.46 | 1.85 | 3.00 | – |
| Potassium (K$_2$O) [g kg DM$^{-1}$] | 11.9 | 11.8 | 12.8 | 12.3 | 11.9 | 13.6 | 16.7 | 11.9 | 11.6 | – |
| Boron (B) [g kg DM$^{-1}$] | 0.13 | 0.12 | 0.13 | 0.16 | 0.15 | 0.15 | 0.36 | 0.29 | 0.32 | – |
| Salmonella cfu [100 g$^{-1}$] | | | | | not found | | | | | 0 |
| Number of living eggs of gut parasites [number kg DM$^{-1}$] | | | | | not found | | | | | 0 |

### 3.2. Effects of Soil Application of Digestate on Maize Yield

Figures 4–6 present the effects of the application of sugar beet pulp digestate (in three forms) on the yield of the studied crop.

An analysis of variance of the average cob weight results found statistically significant differences between the studied soil amendments. The average weight of maize cob harvested from an NPK (control) plot was lower compared with the average weight of the cobs grown on plots with soil amended with either form of the digestate: liquid fraction, unseparated and solid fraction (by 7.1%, 17.2% and 9.7%, respectively) (Figure 4). The cob weight was also significantly determined by the growing season, and the season of 2014 produced cobs with higher weights compared with the two other seasons. It is worth mentioning that differences between experimental treatments in 2015 and 2016 were more pronounced than in 2014.

When analyzing the weight of stover (without cobs), it was found that plants grown on control plots (with NPK) produced significantly lower biomass of stover compared with the biomass obtained from digestate-amended plots, regardless of the digestate form applied. No differences in the weight of stover biomass were found between the plots with different forms of digestate applied to soil (Figure 5). The seasons of the study affected the weight of stover to a high extent, and the season of 2014 recorded the highest stover biomass weights, irrespective of fertilization treatment.

The average weight of above-ground biomass from a single maize plant grown on a control plot was significantly the lowest (Figure 6), whereas the digestate application did not affect this value. Differences in above-ground biomass of maize between the growing seasons were found to be statistically significant, and a higher weight of above-ground biomass was noted in the season of 2015 compared to that in the other two seasons of the study.

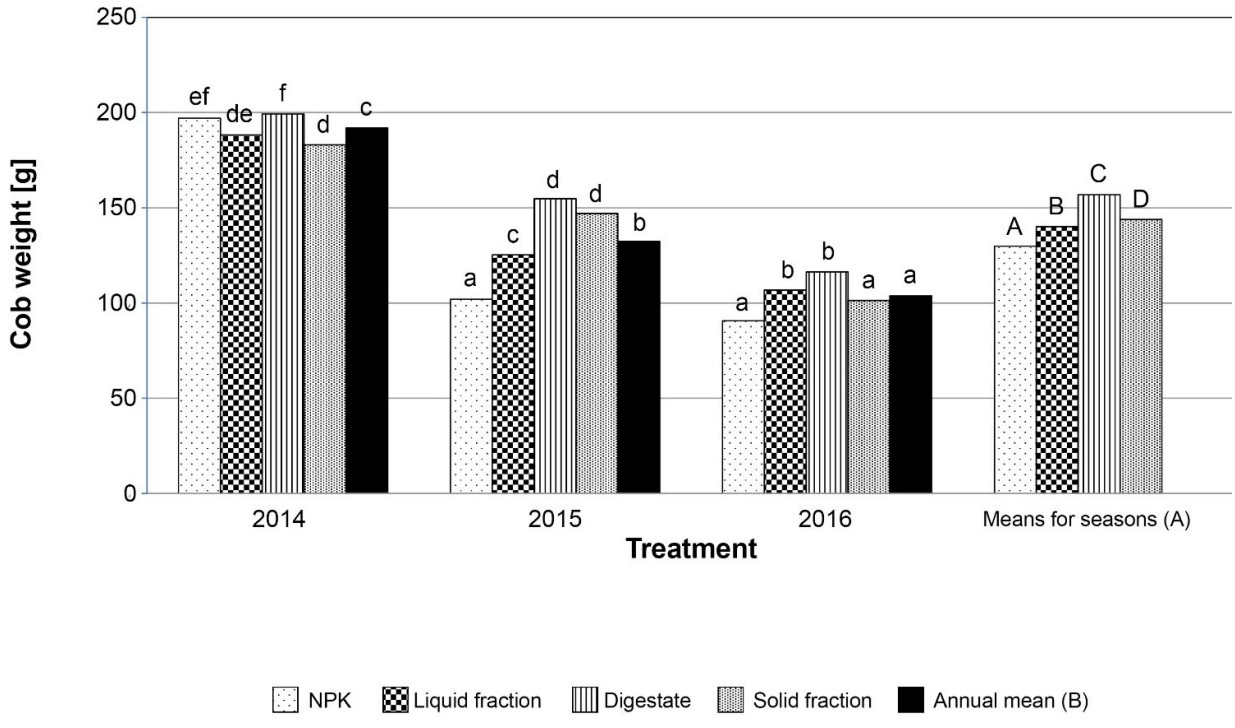

**Figure 4.** Weight of maize cobs in respective growing seasons and experimental treatments. **Legends:** No letter means no significant differences. The same lowercase letter means no significant differences among mean values for a treatment within the same season and for interaction of season × treatment. The same capital letter means no significant differences among the treatments.

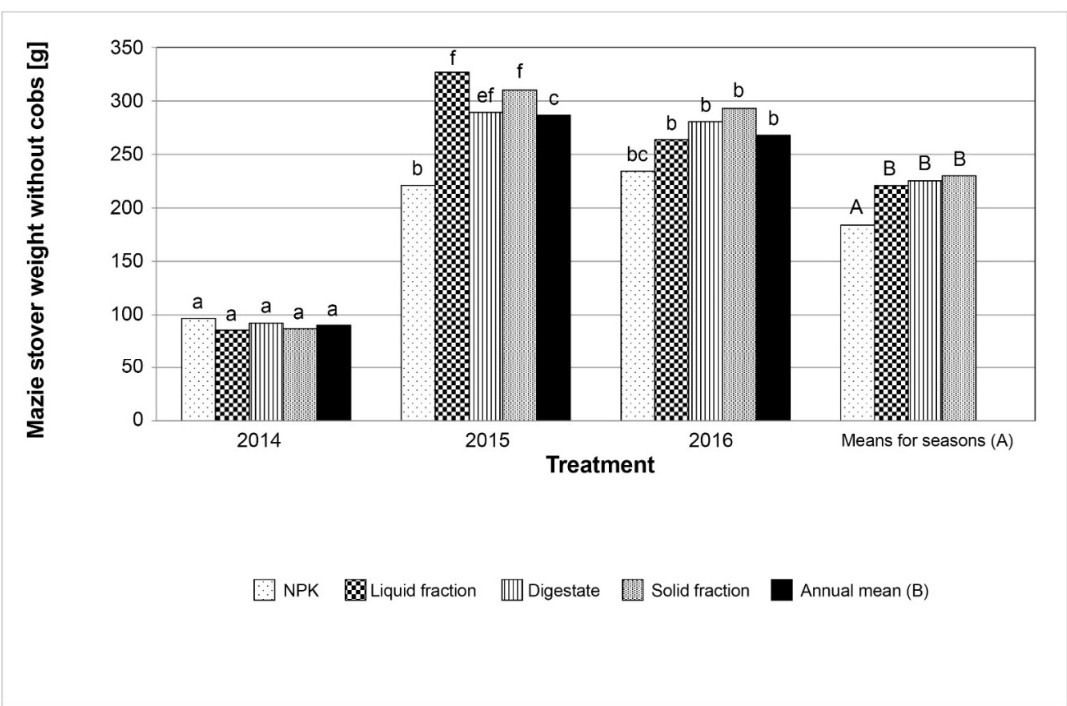

**Figure 5.** Weight of maize stover in respective growing seasons and experimental treatments. Legend as at Figure 4.

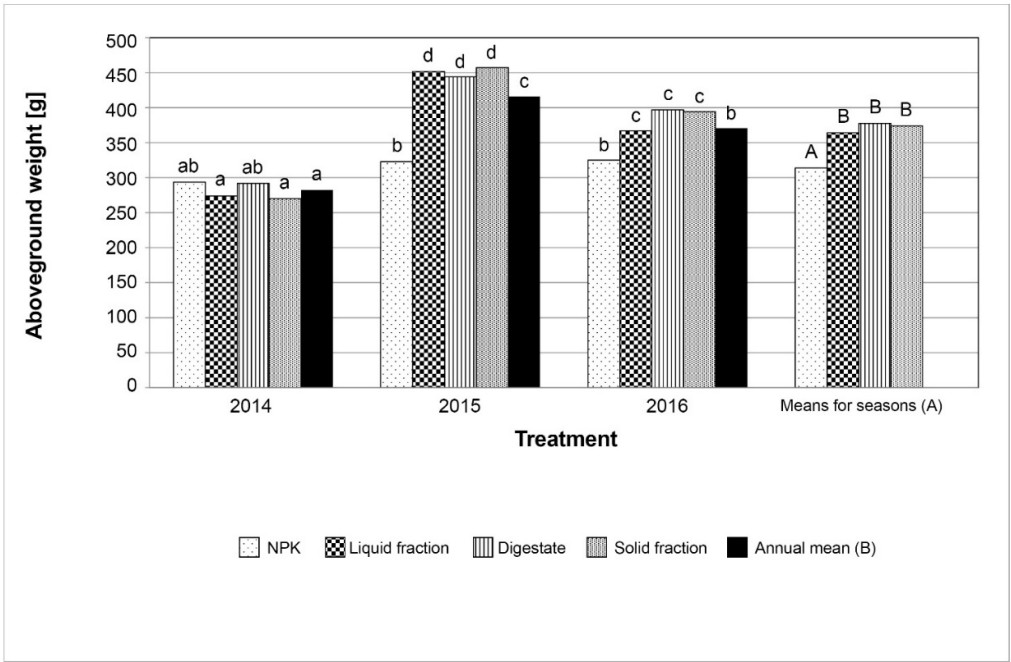

**Figure 6.** Weight of maize above-ground biomass in respective growing seasons and experimental treatments. Legend as at Figure 4.

To sum up, it can be stated that the yield of above-ground biomass and its components (weight of cobs and stover) was positively influenced by digestate application when compared with the yield obtained following standard NPK fertilization. This leads to the conclusion that digestate from the anaerobic digestion of sugar beet pulp can also be effectively applied in the production of crops other than sugar beet.

Although the results of plot experiments cannot be directly extrapolated to the yields obtained under regular farming, the yields calculated per 1 ha plots display the potential of the digestate as a soil amendment in combination with mineral fertilizers.

For NPK-treated plots, the yields of cobs, stover and above-ground biomass were 7.6, 10.8 and 18.4 Mg ha$^{-1}$, respectively. The values for liquid-digestate-treated plots were 8.2, 13.2 and 21.4 Mg ha$^{-1}$ for cobs, stover and above-ground biomass, respectively. The yields obtained from crude-digestate-amended plots amounted to 9.2, 13.0 and 22.2 Mg ha$^{-1}$, while from liquid-digestate-supplemented plots these values were 8.5, 13.5 and 22.0 Mg ha$^{-1}$ for cobs, stover and above-ground biomass, respectively.

The presented data of the predicted yield are in accordance with Polish literature. For example, Gorzelany et al. [34] reported yield ranges of 5–10 and 8–20 Mg ha$^{-1}$ for cob and above-ground biomass, respectively. High yields of above-ground biomass of maize of 15–25 Mg ha$^{-1}$ come in for increased interest in its utilization for energy generation—through silage anaerobic digestion, processing of infested grains to bioethanol or direct combustion of whole plants [28,32].

Since the authors of this study were unable to find reports on the utilization of sugar beet pulp digestate in maize production, the discussion of the results is rather limited.

The effect of digestate application on the yield and quality of crops has not yet been extensively defined.

Gunnarsson et al. [12] reported that digestate application in soil can replace industrial fertilizers only to a limited extent. Alburquerque et al. [42] studied the yielding ability of some vegetable species and found that the effect of digestate and fertilizers can be comparable only for spring crops. For winter crops, the application of industrial fertilizers produced high yields. Vaneeckhaute et al. [43] studied the option of replacing fertilizers with digestate or digestate in combination with manure and reported significantly higher yields. However, maize grown under the conditions of Sweden [44] yielded at the same level irrespective of soil amendment type (digestate or NPK) or soil amendment combination (digestate plus NPK and other treatments). Bueno et al. [45] studied the application of digestate from the anaerobic digestion of maize biomass in a greenhouse trial. Those authors could not confirm differences between the yields of maize cultivated in minerally fertilized soil and digestate-amended soil and concluded that digestate can be effectively used in maize biomass production. Hupfauf et al. [46] determined the effects of digestate application on the growth and yield of several crops (excluding maize) and found that plants grown in digestate-amended soil displayed higher efficiencies compared with crops grown in fertilized soil. Akhiar et al. [40] reported data concerning distribution of nutrients in the liquid and solid fraction of digestates of different origin. Generally, they found that liquid fractions in almost all cases were more abundant and were more suitable as soil amendments than solid fractions. In the case of the digestate reported herein, we found contrary relations. In the authors' opinion it was because the digestate from sugar beet pulp was pre-treated with flocculants and centrifuged, which resulted in a higher concentration of nutrients compared to that in the liquid fraction.

It should be mentioned that under the conditions of the reported experiment, it was found that the digestate can replace mineral fertilizers and this practice can bring economic benefits since the cost of fertilizers of 1500 PLN ha$^{-1}$ can be eliminated (1 € ≈ 4.50 PLN) [47].

### 3.3. Effect of Soil Amendment with Digestate on Chemical Properties of Maize Biomass Yield

In three growing seasons (from 2014 to 2016), maize biomass was harvested, and chemical analyses were performed. Biomass was analyzed in raw form, i.e., it was assumed that the biomass would be taken directly just after harvest to the incineration plant.

Although all determined chemical parameters had an impact on the value of maize biomass as fuel, the contents of volatile compounds and carbon seem to be the most important.

Table 3 presents the contents of the volatile fraction and carbon in maize cobs and, based on these results, it could be concluded that the digestate application in soil resulted

in higher values of both parameters for the unseparated digestate, and this difference was statistically significant. When analyzing maize stover, a significant reduction in the values of both mentioned parameters compared with the control fertilizer was found.

**Table 3.** Effects of fertilization on chemical properties of maize biomass *.

| Treatment (A) | Season (B) | | | Mean (A) | Season (B) | | | Mean (A) |
|---|---|---|---|---|---|---|---|---|
| | 2014 | 2015 | 2016 | | 2014 | 2015 | 2016 | |
| | Maize Cob | | | | Maize Stover | | | |
| | Total moisture, % | | | | | | | |
| Mineral nitrogen | 43.6 [b] | 39.3 [b] | 31.5 [a] | 38.6 | 27.5 [a] | 38.6 [b] | 39.1 [b] | 35.0 [A] |
| Liquid fraction | 40.8 [b] | 39.4 [b] | 29.8 [a] | 37.7 | 51.0 [d] | 47.9 [c,d] | 51.7 [d] | 50.2 [C] |
| Unseparated digestate | 42.9 [b] | 39.0 [b] | 31.2 [a] | 36.7 | 39.6 [b,c] | 42.0 [b,c] | 51.4 [d] | 44.3 [B] |
| Solid fraction | 43.7 [b] | 40.3 [b] | 31.9 [a] | 38.1 | 39.6 [b,c] | 48.2 [c,d] | 52.1 [d] | 46.6 [B,C] |
| Mean value (B) | 42.8 [b] | 39.5 [b] | 31.1 [a] | | 39.4 [a] | 44.2 [b] | 48.6 [c] | |
| | Ash, % | | | | | | | |
| Mineral nitrogen | 1.8 [d] | 1.0 [a] | 1.4 [c] | 1.4 [B] | 6.6 [c,d] | 4.7 [b] | 5.6 [c] | 5.6 [B] |
| Liquid fraction | 1.3 [b,c] | 1.1 [b] | 1.2 [b] | 1.2 [A] | 13.9 [f] | 8.2 [d] | 9.9 [e] | 10.6 [D] |
| Unseparated digestate | 1.3 [b,c] | 1.2 [b] | 1.3 [b,c] | 1.3 [A] | 5.0 [b] | 3.9 [a] | 4.5 [b] | 4.5 [A] |
| Solid fraction | 1.2 [b] | 1.1 [b] | 1.3 [b,c] | 1.2 [A] | 9.1 [e] | 4.6 [b] | 6.8 [b,c] | 6.8 [C] |
| Mean value (B) | 1.4 [b] | 1.1 [a] | 1.3 [b] | | 8.7 [c] | 5.3 [a] | 6.7 [b] | |
| | Volatile parts, % | | | | | | | |
| Mineral nitrogen | 55.9 [c] | 44.5 [a] | 50.5 [b] | 50.3 [A] | 51.5 [d] | 40.9 [c] | 46.4 [c,d] | 46.3 [D] |
| Liquid fraction | 57.1 [c] | 45.4 [a] | 51.4 [b] | 51.3 [A,B] | 27.9 [a] | 32.7 [b] | 30.5 [b] | 30.4 [A] |
| Unseparated digestate | 57.8 [c] | 45.8 [a] | 51.8 [b] | 51.8 [B] | 43.4 [c] | 35.9 [b,c] | 39.8 [c] | 39.7 [B] |
| Solid fraction | 56.9 [c] | 45.0 [a] | 51.2 [b] | 51.0 [A,B] | 32.7 [b] | 35.2 [b,c] | 33.9 [b,c] | 33.9 [C] |
| Mean value (B) | 56.9 [c] | 45.2 [a] | 51.0 [b] | | 38.9 [b] | 36.2 [a] | 37.7 [a] | |
| | Carbon, % | | | | | | | |
| Mineral nitrogen | 31.7 [d] | 25.7 [a] | 28.8 [c] | 28.7 [A] | 32.7 [d] | 26.3 [b,c] | 29.6 [b,c] | 29.5 [C] |
| Liquid fraction | 32.4 [d,e] | 26.3 [a,b] | 29.5 [b,c] | 29.4 [A,B] | 18.0 [a] | 22.4 [a,b] | 20.3 [a] | 20.2 [A] |
| Unseparated digestate | 33.1 [e] | 26.9 [b] | 30.1 [b,c] | 30.0 [B] | 27.4 [b] | 23.7 [b] | 25.7 [b] | 25.6 [B] |
| Solid fraction | 32.4 [d,e] | 26.4 [a,b] | 29.5 [b,c] | 29.4 [A,B] | 20.0 [a] | 23.2 [a,b] | 21.7 [a,b] | 21.6 [A] |
| Mean value (B) | 32.4 [c] | 26.3 [a] | 29.5 [b] | | 24.5 | 23.9 | 24.3 | |
| | Hydrogen, % | | | | | | | |
| Mineral nitrogen | 4.3 [b] | 3.5 [a] | 3.9 [a,b] | 3.9 | 4.2 [d] | 3.2 [b,c] | 3.7 [c] | 3.7 [C] |
| Liquid fraction | 4.4 [c] | 3.6 [a] | 4.1 [b] | 4.1 | 2.2 [a] | 2.7 [a,b] | 2.5 [a] | 2.5 [A] |
| Unseparated digestate | 4.5 [c] | 3.7 [a] | 4.2 [b] | 4.1 | 3.4 [b,c] | 2.9 [b,c] | 3.1 [b,c] | 3.1 [B] |
| Solid fraction | 4.4 [c] | 3.6 [a] | 4.2 [b] | 4.1 | 2.4 [a,b] | 2.8 [b,c] | 2.6 [a] | 2.6 [A] |
| Mean value (B) | 4.4 [c] | 3.6 [a] | 4.1 [b] | | 3.0 | 2.9 | 3.0 | |
| | Nitrogen, % | | | | | | | |
| Mineral nitrogen | 0.68 [a] | 0.82 [b] | 0.76 [a,b] | 0.75 [A] | 0.43 [d] | 0.30 [b] | 0.38 [c] | 0.37 [C] |
| Liquid fraction | 0.72 [a] | 0.89 [c] | 0.82 [c] | 0.81 [B] | 0.23 [a] | 0.29 [a,b] | 0.27 [a,b] | 0.26 [A] |
| Unseparated digestate | 0.78 [a,b] | 1.05 [e] | 0.88 [c,d] | 0.91 [B] | 0.35 [b,c] | 0.29 [a,b] | 0.32 [b] | 0.32 [B] |
| Solid fraction | 0.74 [a] | 0.95 [d] | 0.85 [c] | 0.85 [B] | 0.28 [a,b] | 0.31 [b] | 0.31 [b] | 0.30 [A,B] |
| Mean value (B) | 0.73 [a] | 0.93 [c] | 0.83 [b] | | 0.30 | 0.32 | 0.32 | |
| | Sulphur, % | | | | | | | |
| Mineral nitrogen | 0.06 [a] | 0.07 [a,b] | 0.08 [a,b] | 0.07 [A] | 0.06 [b] | 0.04 [a,b] | 0.06 [b] | 0.05 |
| Liquid fraction | 0.07 [a,b] | 0.07 [a,b] | 0.08 [a,b] | 0.07 [A] | 0.04 [a,b] | 0.04 [a,b] | 0.05 [a,b] | 0.04 |
| Unseparated digestate | 0.07 [a,b] | 0.08 [a,b] | 0.09 [b] | 0.08 [B] | 0.05 [a,b] | 0.03 [a] | 0.05 [a,b] | 0.04 |
| Solid fraction | 0.07 [a,b] | 0.07 [a,b] | 0.08 [a,b] | 0.07 [A] | 0.04 [a,b] | 0.03 [a] | 0.05 [a,b] | 0.04 |
| Mean value (B) | 0.07 [a] | 0.07 [a] | 0.08 [b] | | 0.05 [b] | 0.04 [a] | 0.05 [b] | |

* No letter means no significant differences. The same lowercase letter means no significant differences among mean values for a treatment within the same season and for interaction of season × treatment. The same capital letter means no significant differences among the treatments.

No relationships were found between the water and hydrogen contents in maize cobs and the studied soil treatments. It was found that cobs harvested from control plots displayed a significantly higher ash content compared with cobs cultivated in soil amended

with digestate. For nitrogen content in cobs, the maize's response to the experimental treatments was the opposite.

When analyzing the results obtained for maize stover, it can be concluded that the moisture content in maize from control plots was significantly lower than in plants harvested from all plots treated with digestate, whereas the contents of hydrogen and nitrogen were the highest in stover from control plots.

To sum up, the differences in the chemical composition of maize biomass (cobs and stover) obtained from different experimental plots are relatively small and the present study proved the suitability of the digestate application in soil used for the cultivation of maize for energy purposes.

### 3.4. Effects of Soil Application of Digestate from Anaerobic Digestion of Sugar Beet Pulp on Energy Value of Maize Biomass Yield

Results of effects of soil application of digestate (in three forms) on energy parameters of maize biomass are presented below (Figures 7–10).

When analyzing the energy value of undried maize cobs determined directly after harvest, it was found that soil amendment with raw digestate resulted in significantly higher gross and net energy values compared with other studied treatments (Figures 7 and 8). The difference between gross and net energy values consists of energy used for water evaporation. It is worth mentioning that the differences between energy values of maize cobs obtained in the respective experimental seasons of the study were also found to be statistically significant, and the highest cob energy values were recorded in 2014 and the lowest in 2015.

While considering the energy value of the combustion of maize stover (Figures 9 and 10), the treatment type and study season were found to have a significant impact. Similar to the energy value of maize cob, the highest gross and net energy values were recorded in the season of 2014. The effect of the treatment type on the energy value of stover was also statistically significant and stover from the control plot (NPK) displayed the highest values of heat energy compared with stover from digestate-amended plots.

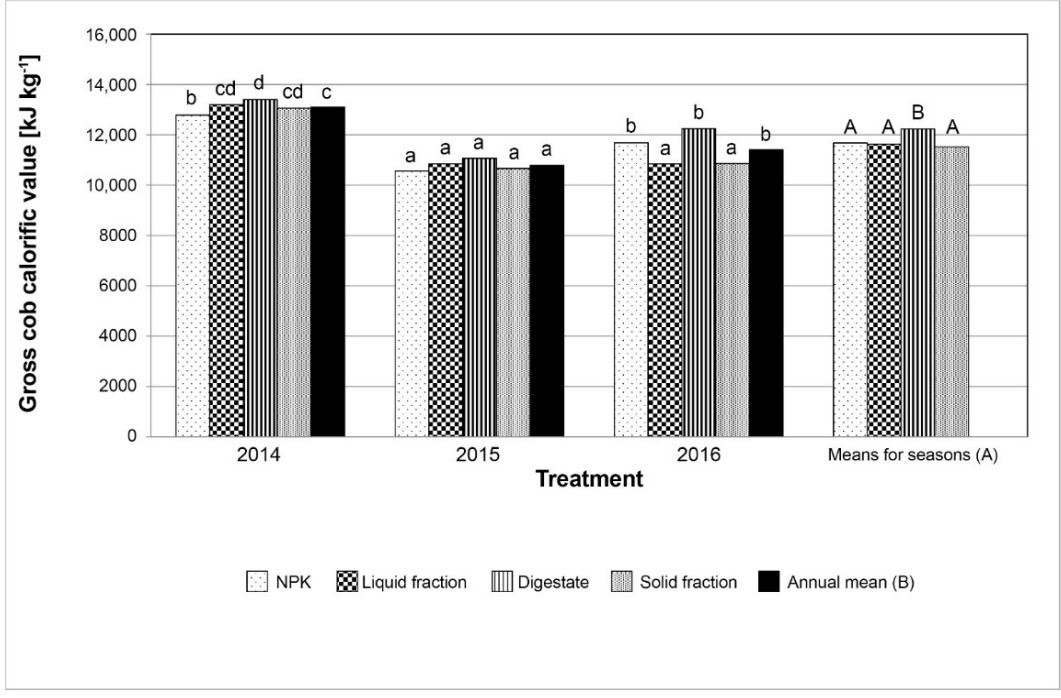

**Figure 7.** Gross energy value of maize cobs (as a raw material) in respective growing seasons and studied treatments. Legend as at Figure 4.

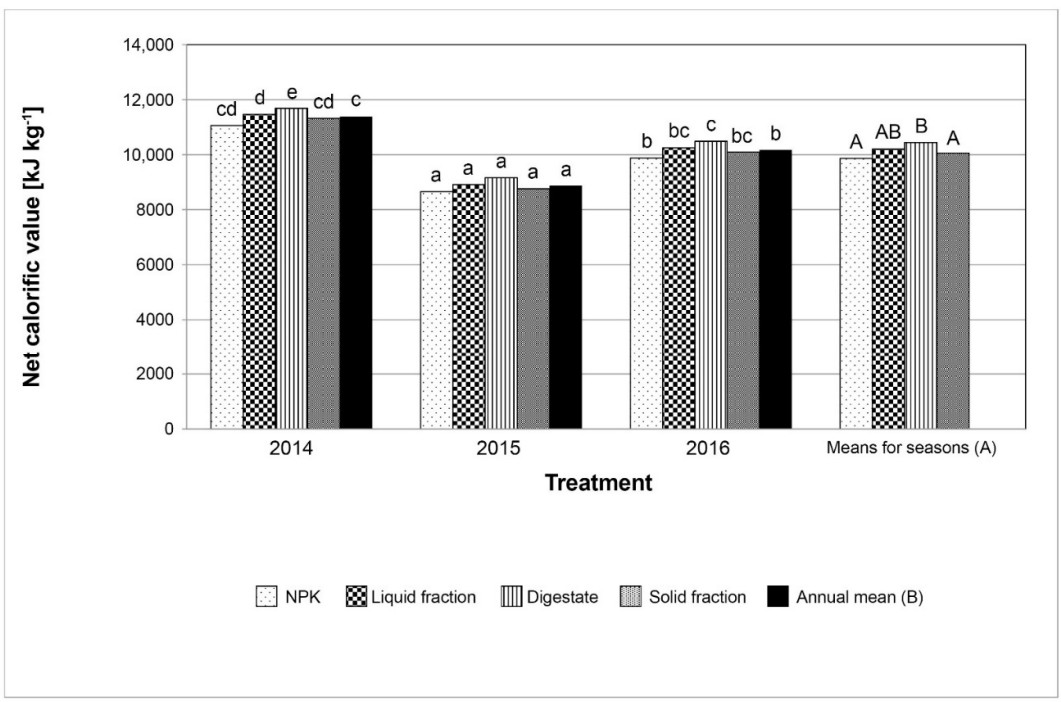

**Figure 8.** Net energy value of maize cobs (as a raw material) in respective growing season and studied treatments. Legend as at Figure 4.

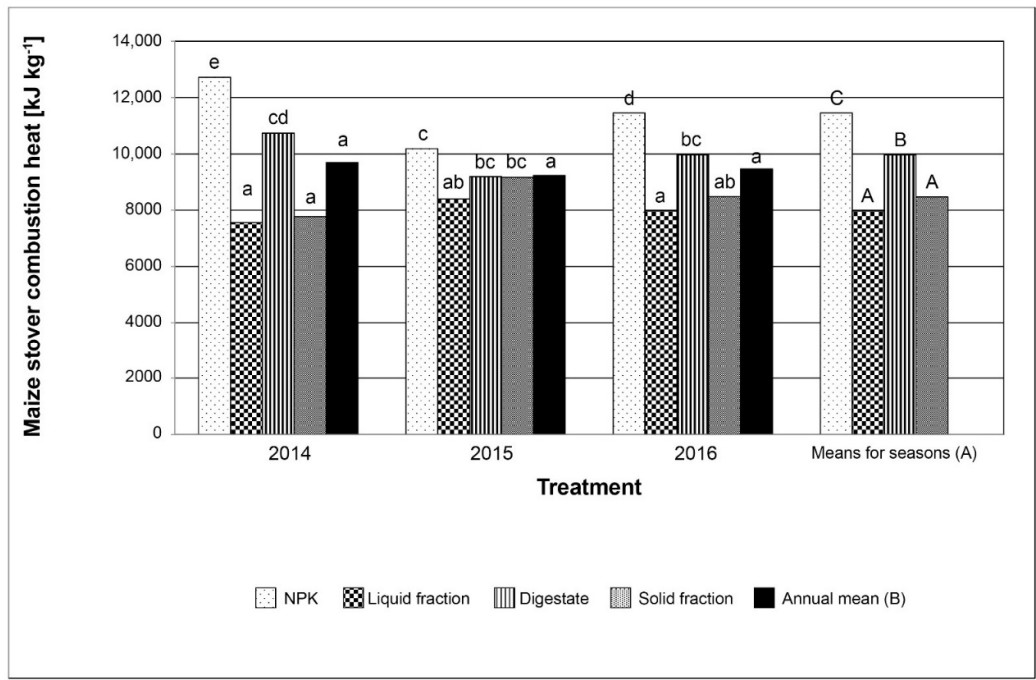

**Figure 9.** Gross energy value of maize stover (as a raw material) in respective growing seasons and studied treatments. Legend as at Figure 4.

Since the gross calorific value of lignite and wood is 9–12 and 7–12 MJ kg$^{-1}$, respectively, and the respective calorific values for cobs and stover recorded in the present study are approximate to the upper limit for cobs and the lower limit for stover, it can be concluded that maize is an equally good source of heat energy.

A proposed design is presented in Figure 11. According to the authors' knowledge and experience, it presents a concept and method of effective utilization of digestate for the

production of energy maize that can be suitable on farms where sugar beets and maize are grown in crop rotation. It presents a circular-economy attitude and shows the possibility of nutrient recycling in the framework of a farm specialized in sugar beet growing.

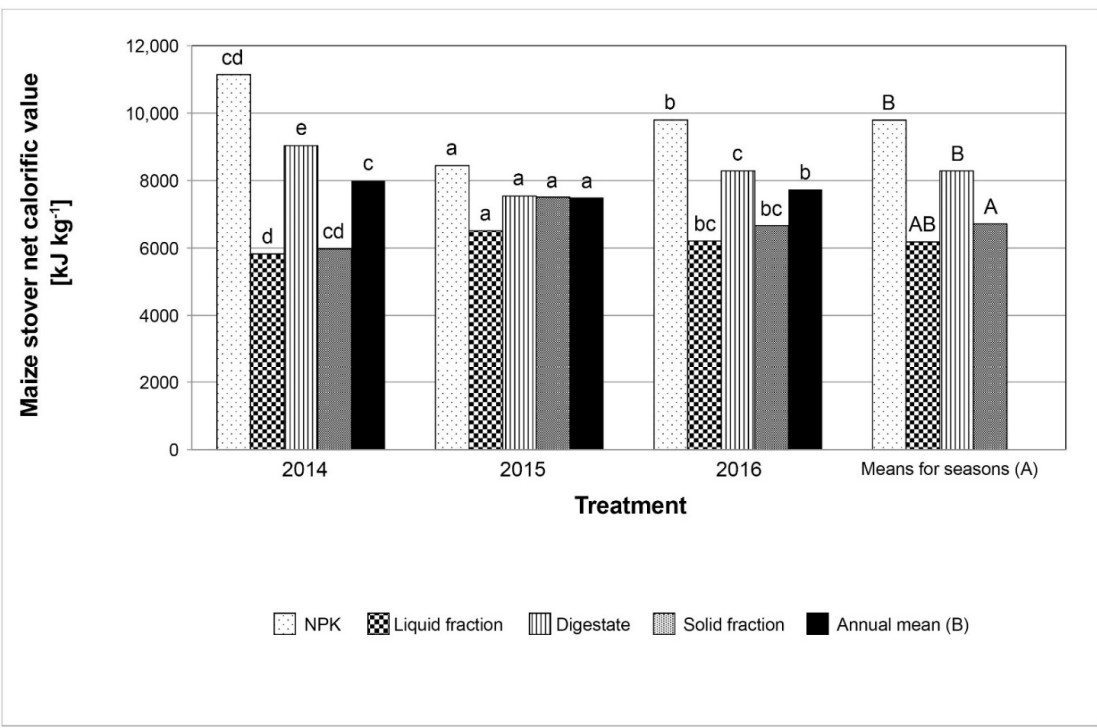

**Figure 10.** Net energy value of maize stover (as a raw material) in respective growing seasons and studied treatments. Legend as at Figure 4.

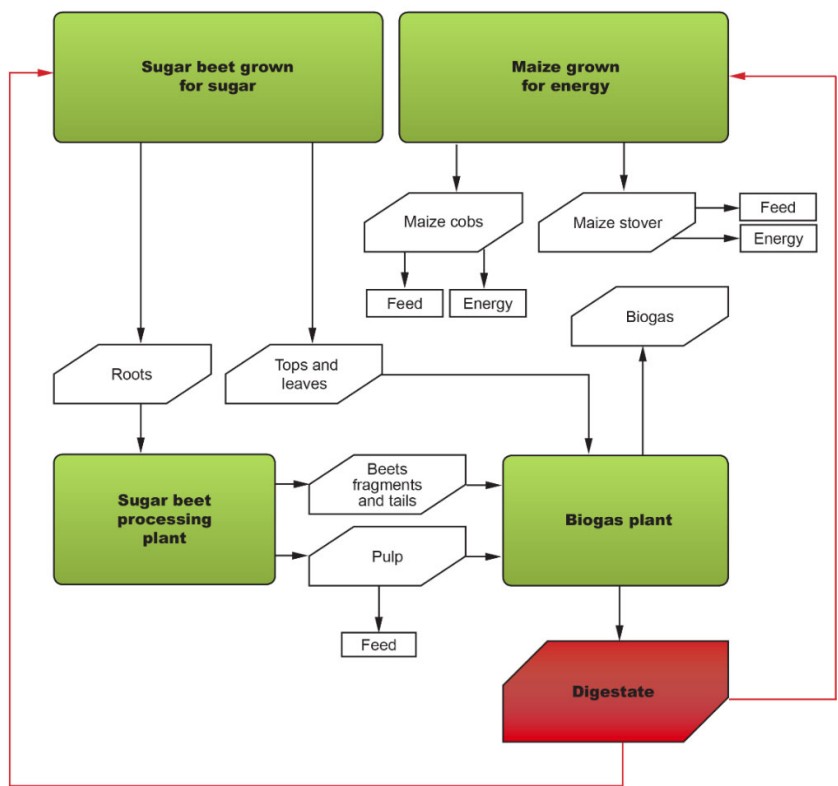

**Figure 11.** Design of closed-system digestate utilization (by authors).

## 4. Conclusions

1.  The studied by-products of sugar beet pulp anaerobic digestion (raw digestate, liquid and solid digestate fractions) can be effectively applied to soil used for growing maize for energy purposes.
2.  The solid fraction of the studied digestate was a better option for application as far as yield-bearing effects are concerned. This was due to the physical nature of this fraction and pretreatment including flocculation.
3.  The yields of maize cobs and stover were affected by the studied season and soil treatment, and the yields of cobs and stover harvested from plots with digestate-amended soil were higher than the yields of these plant components obtained from control NPK plots.
4.  Energy values of maize biomass confirmed that the digestate application to soil is an option generating high amounts of heat energy with the utilization of waste by-product.
5.  Replacing mineral fertilizers with digestate can bring economic benefits since the cost of mineral fertilization in Poland is 1500 PLN per 1 ha (1 € ≈ 4.50 PLN).

**Author Contributions:** Conceptualization, A.B.; methodology, A.B. and B.P.; software, A.K.; validation, T.P.O. and A.B.; formal analysis, A.K. and A.B.; writing—original draft preparation, A.B. and B.P.; writing—review and editing, A.K. and T.P.O. All authors have read and agreed to the published version of the manuscript.

**Funding:** This research received no external funding

**Conflicts of Interest:** The authors declare no conflict of interest.

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
