# Peer review of "Application of Sugar Beet Pulp Digestate as a Soil Amendment in the Production of Energy Maize"

_processes, doi:10.3390/pr9050765_

Round 1

Reviewer 1 Report

Thank you for submitting this work to processes.
Below a few comments on how I think you can improve this manuscript.

-Abstract: can you quantify the better results - that would be great
-I guess you could also substitute mineral fertilizers that have potentially hazardous chemicals such as U - there is a recent study from a Polish researcher on this (https://doi.org/10.1016/j.resourpol.2019.02.012)
-Figure 11, the general layout of the proposed process you could already present in the intrpduction - I think this could help

This is a fine manuscript with great experiments.
I recommend english editing - apart from that I think this is pretty much ready for publication
I find the results very interesting and promising

Author Response

Dear Reviewer, Thank you for your review. Please note that your all of your remarks have been takenm into account.  Revision of English languagewas made by native English speaker who made a lot of corrections and also suggested changing title of the mmanuscript. Authors have agreed for that. Abstract was modigfied - lines 35 and 36. New citation was introduced lines 69-72 and reference no 19. modification in introduction was made lines 122-124 according to Reviewer's remark.

Reviewer 2 Report

Comment on manuscript No. Processes-1166984: Suitability of soil application of digestate obtained from the digestion of sugar beet pulp in the production of maize for energy purposes

Manuscript No. Processes-1166984 is a valuable study for assessing the application of sugar pulp digestate for improving the Zea mays production. Authors have studied the yield and energy benefits simultaneously for the applied treatments, however, the study could have been technically sound if they have explained this with proper rationale, detailed methodology, appropriate representation of results, logical discussion, and some numeric conclusion. After adding aforesaid information, the present manuscript could be suitable for publication in Processes Journal of MDPI. Authors are requested to address the following comment during the revision of the present manuscript.

  1. Kindly mention the sources of the sugar pulp digestate in 3-4 sentences. 
  1. I suggest a table for the individual treatments in the material and method section of the MS. 
  1. The author should add the experimental design by explaining how the experimental plots and treatments were adopted throughout the study. 
  1. Please modify the writing style of the units in table 1, it’s better to arrange them into the same column where parameters have been mentioned. 
  1. Kindly change the highlight the legends and parameters in all the figures starting from 4 to 10, as it is difficult to read. 
  1. In the discussion, kindly explain the doses of treatment and compare your findings and their efficacy, cost-effectiveness, and sustainability with the other studies. Also, provide numerical data for the same where applicable.   
  1. Elaborate the caption of figure 11 and explain the process based on the findings.   
  1. Kindly mention, which fraction of un-separated digestate, liquid and solid is the most suitable fraction for application and discuss the same with the finding of the present study.
  1. In the conclusion, kindly provide the to the point (which fraction) information of digestate application with data of the present study      

Author Response

Dear Reviewer, Thank you for your efforts and detailed review of our manuscript. Please note that manuscript was revised by native English speaker and that he suggested to modify title of our manuscript and the authors have agreed for his modifications. Source of digestate was given in lines: 130-134. Table was inrtoduced as Table 1 line 145. Design of experimental plots was given 136-138. It was the typical design: Control, 1, 2, 3; 1, Control, 3,2; 2, 3., Control, 1 and the last row 3, 2,1, Control. I do trust that that statement of Latin square is sufficient. Table 1 (now Table 2) has been re-designed according to Reviewers' suggestion. Figures 4 to 10 were edited according to Reviewers' suggestion. Line 304-310 were added with new reference no 40 Akhiar et al. (2017) according to Reviewers' suggestion. New coclusion no 2 was added.